
# Influence of Geant4 physics list
# on simulation accuracy and performance

Róbert Breier[1], Alexander Fuss[2,3], Holger Kluck[2], Valentyna Mokina[2],
Veronika Palušová[4*] and Pavel Povinec[1]

**1** Department of Nuclear Physics and Biophysics, Faculty of Mathematics, Physics and
Informatics, Comenius University, 84248 Bratislava, Slovakia
**2** Institut für Hochenergiephysik der Österreichischen Akademie der Wissenschaften,
Wien, 1050, Austria
**3** Atominstitut, Technische Universität Wien, Wien, 1020, Austria
**4** Institute of Experimental and Applied Physics, Czech Technical University in Prague,
Prague, CZ-11000, Czech Republic

⋆ veronika.palusova@cvut.cz

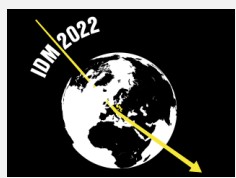

## Abstract

**A main goal of current low background physics is the search for rare and novel phenomena beyond the Standard Model of particle physics, e.g. the scattering off of a potential Dark Matter particle inside a CaWO$_4$ crystal or the neutrinoless double beta decay of Ge nucleus. The success of such searches depends on a reliable background prediction via Monte Carlo simulations. A widely used toolkit to construct these simulations is GEANT4, which offers a wide choice of physics models, so-called physics lists. To facilitate the selection of physics lists for simulations of CaWO$_4$ and Ge targets, we quantify their impact on the total energy deposition for several test cases.**


## 1 Introduction

Rare event searches, such as the search for dark matter or neutrinoless double beta decay, depend crucially on a reliable and verified background prediction. Commonly, these predictions are based on Monte Carlo simulations of the relevant background sources. A widely used toolkit to create these simulations is GEANT4 [1–3].

In GEANT4, the physics models that describe the particle processes are specified in the so called *physics list*. Albeit the user is generally free to specify their own collection of models,

GEANT4 provides with *physics constructors* predefined collections of processes to cover related particle processes that can be included in the physics list. In this work we will focus on the *electromagnetic* physics constructors.

In literature, the electromagnetic physics constructors and their settings are validated for a wide range of physics processes and observables, e.g. electron back scattering [4]. However, to our knowledge, no such study exist about their impact on the total energy deposited by radioactive decays in CaWO$_4$ or Ge target material. As pointed out in [4], an observable like energy deposition by radioactive decays is the result of several physics processes. Hence, it can only be assessed for a specific use case, and it is usually precarious to extrapolate it from studies based on a different use case.

With this work we provide first results of a dedicated study for this missing use case: examine the impact of different electromagnetic physics constructors on the total energy deposition for several test cases, i.e. combinations of radioactive contaminants, target material (CaWO$_4$ and Ge) and target thickness. The goal is to give an assessment to what extent the selected physics constructors affect the simulated observable of total energy deposition, i.e. how compatible different physics constructors are. To assess the compatibility in a qualitative and objective way, we adopt the methodology established in [4–7]. To give some guidance which physics constructor to choose in case of compatibility, we consider also the computing performance.

## 2 Methodology

The strategy of this study is focused on the simulation of total energy deposition in a bulk crystal of two target materials, i.e. CaWO$_4$ and Ge. Simulations were produced with the ImpCRESST physics simulation code [8] based on GEANT4 version 10.6.3 released in 2020. *Test cases*, characterized by target material, target thickness and radioactive contaminant, are simulated for several GEANT4 physics constructor. The impact of different physics constructors on the total energy deposition in our test cases is assessed by the means of statistical methods.

### 2.1 Physics Constructor Configurations

GEANT4 provides several predefined physics constructors for electromagnetic (EM) physics. Twelve of them were used during this work: G4EmStandardPhysics_option1, 2, 3 and 4, G4EmLivermorePhysics, G4EmLivermorePolarizedPhysics, G4EmPenelopePhysics, G4EmStandardPhysics, G4EmStandardPhysicsGS, G4EmLowEPPhysics, G4EmStandardPhysicsWVI, and G4Em-StandarPhysicsSS.

We treat each physics constructor as its own one parametric-model with the range cut for secondary particle production as a free parameter. In a GEANT4 simulation, secondary particles that are unable to travel further than the range cut value, called *production cut*, are not produced, but their energy is deposited locally [9, p.256]. In order to study the impact of such parameter and to possibly improve the performance of simulations, each simulation was performed for five production cut values: 100 nm, 1 μm, 1 mm, 1 cm , 10 cm. In this work, one GEANT4 physics constructor *configuration* is characterized by physics constructor and cut value, yielding together 60 configurations (=12 physics constructors × 5 production cut values).

### 2.2 Test Cases

This study covers two target materials - CaWO$_4$ or Ge. The target geometry was defined as a cuboid with a cross-section of 32 × 32 mm in two configurations: *bulky* with a thickness

of 32 mm and *thin* with a thickness of 50 μm.[1] Six common radioactive contaminants in chosen targets are considered - low Q-value $\beta$ emitters ($^{228}$Ra, $^{210}$Pb), high Q-value $\beta$ emitters ($^{208}$Tl, $^{210}$Tl) and $\alpha$ emitters ($^{211}$Bi, $^{234}$U).

Each one of the 24 test cases (= 2 targets × 2 thicknesses × 6 contaminants) is simulated for each configuration, yielding together 1440 datasets (= 24 test cases × 60 configurations). As G4EmStandardPhysics_option4 is regarded the most accurate model[2] we chose it together with a 1 mm cut as reference dataset.

## 2.3 Statistical analysis

Quantitative comparisons between simulated test case and reference datasets are determined by statistical tests and performed in two steps. In the first step, we address the question of how well a simulated dataset distribution is described by the distribution of the reference dataset by appropriate goodness-of-fit tests. The second step is based on categorical tests, which determine if the difference in compatibility observed across our GEANT4 configurations is statistically significant. This statistical significance is evaluated in two cases: for simulations using *different* physics constructors, which produce unpaired samples, and for simulations using the *same* physics constructor but differ in a secondary feature, i.e. in production cut, which are related and produce paired samples.

As the statistical tests are not uniformly sensitive to differences in distributions at all values, a variety of statistical tests is applied in each step of analysis to minimize the possibility of introducing systematic effects in validation of results. Three independent goodness-of-fit tests were performed to test compatibility: Anderson-Darling (AD) [10], Kolmogorov-Smirnov (KS) [11, 12] and $\chi^2$ test [13, 14]. The tests used to compare groups of categorical data for a significant difference are $\chi^2$ test of independence and Fisher's exact test [15] for unpaired data, and McNemar's test [16] for paired data. In each test, the significance level $\alpha$ was chosen at 5%, a conventional value.

## 3 Results

We show first results of step one of the statistical analysis, i.e. comparing the relative compatibility of energy deposition of several test cases with the reference configuration. Figure 1 shows the example of energy deposition per single event of $^{210}$Tl decay in CaWO$_4$ for G4EmStandardPhysics_option1 and two cut values compared with the reference configuration. The outcome of $\chi^2$ goodness-of-fit tests is reported in the form of efficiency, which is defined as the fraction of test cases where the p-value resulting from the $\chi^2$ tests is larger than $\alpha$. This quantifies the capability of a configuration to produce statistically consistent results with the reference configuration. Results of the efficiency for GEANT4 physics constructors are shown in Figure 2. This efficiency is plotted for all configurations combined, regardless of the target material and thickness.

The analysis identifies G4EmLivermore as a constructor that is robust against changes of the production cut. Hence, tuning of the production cut may improve the performance of simulations without affecting its outcome. Analysis with AD and KS goodness-of-fit tests, and categorical analysis based on the results of compatibility, are currently conducted.

---

[1]Here, the thicknesses are motivated by the largest and smallest distinct parts in some CRESST detector module [8] (the absorber crystal and a bond wire respectively) to demonstrate extreme cases for the dimensions. However, the target thickness is an experiment specific value and future studies may cover a wider range of thicknesses and target materials, e.g. Si.

[2]See [9, p.215]: "G4EmStandardPhysics_option4, containing the most accurate models from the Standard and Low Energy Electromagnetic physics working groups."

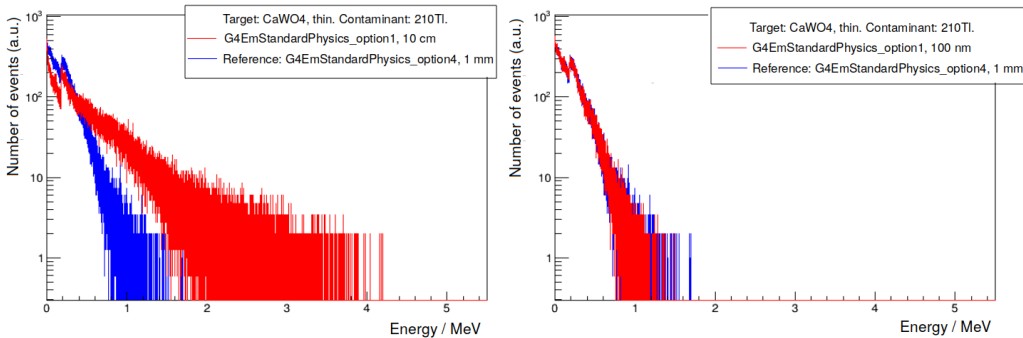

Figure 1: Energy deposition per single event in thin CaWO$_4$ target for 2 configurations. Left - example of failed test of compatibility, right - example of passed test of compatibility.

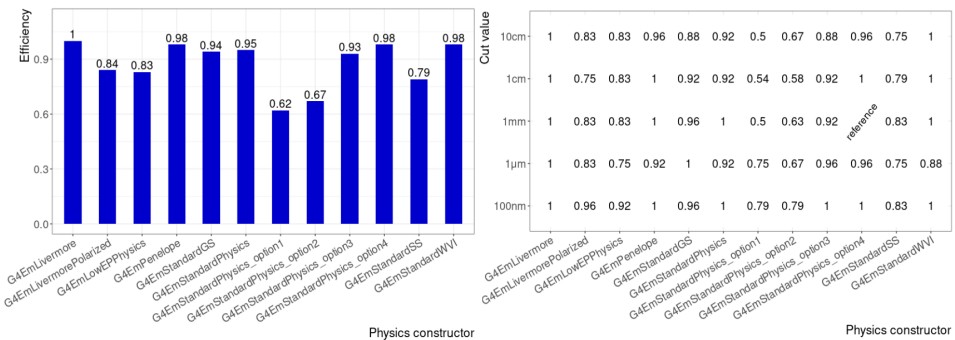

Figure 2: Efficiency of GEANT4 physics constructors (left) and physics constructor configurations (right) based on the results of $\chi^2$ goodness-of-fit test. Efficiencies are calculated regardless of the target material and thickness.

Besides maximizing the accuracy of simulations, one also wants to optimize the computational performance in terms of CPU time. Figure 3 shows the fraction of average run time of GEANT4 configurations relative to the reference dataset. The decreased performance of single scattering models (G4EmStandardWVI and G4EmStandardSS) is expected. Except for the 100 nm case, most values for the production cut affect the relative performance by $\mathcal{O}(10\%)$.

## 4 Conclusion

With this study, we aim to determine the compatibility between total energy deposition of different GEANT4 EM physics constructors compared to the most accurate one, G4EmStandard-Physics_option4. Besides the effects due to the physics constructor, this study also evaluates the sensitivity to features such as the production range cut value. Compared to G4EmStandard-Physics_option4, we found that G4EmLivermore is more efficient in terms of $\chi^2$ goodness-of-fit test and its outcome less dependent on the production cut value.

All configurations of physics constructors feature similar computation performance within $\mathcal{O}(10\%)$ in terms of CPU time, except two cases: configurations with constructors based on single scattering models (i.e. G4EmStandardWVI and G4EmStandardSS) or a production cut value of 100 nm are substantially slower than any other configuration.

First results of this analysis could already provide guidance to users which physics constructor to include in the physics list of their GEANT4 based background prediction studies and how to optimize the configuration of the constructor. Further analysis will be conducted to

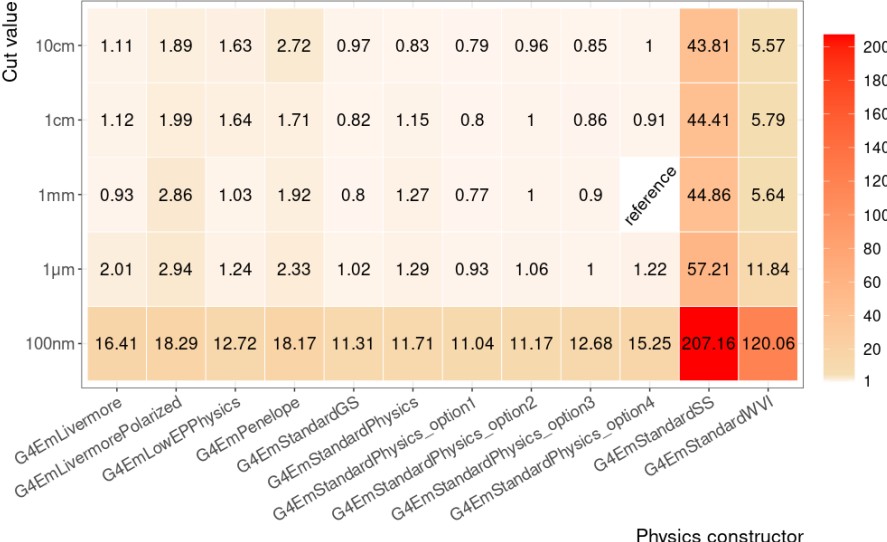

Figure 3: Average run time of GEANT4 physics constructor configurations relative to the reference configuration. Run time is plotted regardless of the target material and thickness.

characterize the differences in simulated total energy depositions observed over the various GEANT4 electromagnetic physics constructors. A publication with the final results is currently under preparation.

## Acknowledgements

**Funding information** This work has been funded through the Sonderforschungsbereich (Collaborative Research Center) SFB1258 'Neutrinos and Dark Matter in Astro- and Particle Physics', by the Austrian science fund (FWF): I5420-N, W1252-N27, P34778-N and by Austria's Agency for Education and Internationalisation (project SK 06/2018). The Bratislava group acknowledges a partial support provided by the Slovak Research and Development Agency (projects SK-AT-2017-001, APVV-15-0576 and APVV-21-0377). This work was supported by the Ministry of Education, Youth and Sports of the Czech Republic under the Contract Number LM2023063.

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
