# Peer review of "Influence of Geant4 Physics List on Simulation Accuracy and Performance"

_SciPost Physics Proceedings, doi:SciPost Phys. Proc. 12, 063 (2023)_

## Round 1 · Referee Report · Anonymous · 2022-10-14

Strengths

1. This kind of comparison is definitely needed.

2. Correct approach for a proper statistical analysis given a large number of simulated datasets.

3. Correct models (physics lists) and parameters identified for comparison.

Weaknesses

1. It would be good to see a brief description of physics lists/constructors to understand the differences between them. I understand that this may significantly increase the length of the paper so I leave this to the authors to decide.

2. Since different particles produce different signals (energy depositions) in detectors and discrimination between these signal is important for identifying dark matter signal, it would be good to see the comparison between different physics lists for specific particle types, like electron recoil (from gammas and betas) and nuclear recoils, and for low energy events as well. This may be a topic for future studies.

3. I think a production cut of 1 cm or higher in the target is too high. Even 1 mm looks too high. So the comparison should focus on cut values around 1 micron, possibly up to 100 microns.

4. In the 'conclusions', the reference configuration is referred to as 'the most accurate model' but there is no proof of this. It may be good to refer to other studies, comparison to experimental data to prove that this is indeed the case.

Report

All acceptance criteria are met. I recommend publication in this journal and suggest some minor changes.

Requested changes

Section 2.1, 3rd line.
Either "were used" or "are used". I prefer "were used".

Section 2.1, 2nd paragraph, 3rd line.
"... to travel further than the range cut value..."

Figure 1. Is the 'thin' target realistic for a dark matter experiment? I'd rather compare effects for the 'thick' target and small production cuts (1 mm and below). Please, explain briefly the difference between the two models.

Figure 2. Is this for thin or thick target?

Figure 3. Can you change the colour for the font, please, on this figure? Black on dark blue is not visible.

Add a brief info at least about the reference case of G4EmStandardPhysics_option4

Check references; some of them miss capital letters in paper titles. The DOI for the last reference looks incomplete.

---

## Round 2 · Referee Report · Anonymous · 2022-12-7

Strengths

Same as in the previous submission

Weaknesses

Mostly same as in the previous submission but some unclear statements have been clarified in the 2nd version.

Report

Criteria met; recommend to publish

Requested changes

Abbreviation for goodness of fit is used in Conclusions but I do not see this to be introduced earlier.

I suggest to say explicitly in Figure 2 and 3 captions which detector size was used. This is not clear.

---

## Round 3 · Referee Report · Anonymous (Referee 1) · 2022-12-14

Strengths

I am happy with the changes.

Weaknesses

N/A.

Report

Recommend to publish

---

## Round 3 · List of Changes

• GoF abbreviation has been changed to goodness-of-fit in the conclusion section.
  • A sentence has been added to text of Section 3 and captions of figures 2 and 3 to say explicitly that the efficiency and run time is calculated and plotted regardless of the size of the detector. A separate analysis and statements about targets (CaWO4 vs Ge) and target thickness (thin vs bulky) will be a part of the upcoming publication with the final results.

---

## Editorial Decision

published